# Autosomal Dominant Polycystic Kidney Disease-Related Multifocal Renal Cell Carcinoma: A Narrative Iconographic Review

**DOI:** 10.3390/ijms26093965

**Published:** 2025-04-23

**Authors:** Consolato M. Sergi, Luis Guerra, Josef Hager

**Affiliations:** 1Anatomic Pathology, Children’s Hospital of Eastern Ontario (CHEO), University of Ottawa, Ottawa, ON K1H 8L1, Canada; 2Pediatric Urology, Children’s Hospital of Eastern Ontario (CHEO), University of Ottawa, Ottawa, ON K1H 8L1, Canada; lguerra@cheo.on.ca; 3Pediatric Surgery, Medical University of Innsbruck, 6020 Innsbruck, Austria; josef.hager@i-med.ac.at

**Keywords:** kidney, classification, cyst, tumor, ADPKD, renal cell carcinoma, NLRP3 inflammasome

## Abstract

Autosomal Dominant Polycystic Kidney Disease (ADPKD) is the most common inheritable disease of cystic degeneration in the kidney. ADPKD is a significant cause of end-stage renal disease (ESRD). Autosomal Dominant Polycystic Liver Disease (ADPLD) results in substantial PLD with minimal PKD. Currently, there are eight genes which have been associated with ADPKD (*PKD1* and *PKD2*), ADPLD (*PRKCSH*, *SEC63*, *LRP5*, *ALG8*, and *SEC61B*), or both (*GANAB*). The severity of ADPKD can show an extremely broad range, but the evolution to ESRD is doubtless unavoidable. In some patients, carcinogenesis develops with inflammation as a potential promoting factor. In this chapter, we illustrate the severity of ADPKD and the fate to develop renal cell carcinoma (RCC).

## 1. Introduction

Autosomal Dominant Polycystic Kidney Disease (ADPKD) is the most common inheritable disease of cystic degeneration in the kidney. Although several classifications have subsequently added complexity to this topic of the upper urinary system, ADPKD remains universally known affecting all populations in all continents. It is unanimously considered an important differential diagnosis in the setting of cystic renal disease worldwide. Potter classification, named after the pediatric pathologist involved in first classifying renal cystic disease, has often been changed into new and different classifications, often based on genetic tests [1]. The ADPKD cysts arise from 1–3% of the nephrons, and they potentially involve all segments of the nephron unit, although most of the cysts are derived from the collecting ducts, which develop from the ureteric bud and are the outgrowth from the caudal end of the Wolffian duct [1,2].

Two genes (*PKD1*, or polycystin 1, and *PKD2*, or polycystin 2) are more often associated with ADPKD, but a third gene (*GANAB*) has been found in a cohort of patients. The *GANAB* (Glucosidase alpha II subunit) gene is a member of the glycosyl hydrolase 31 family of proteins. Glucosidase II plays a role in protein folding and quality control by cleaving glucose residues from immature glycoproteins at the level of the endoplasmic reticulum [3]. *PKD1* encodes a member of the polycystin protein family. The encoded glycoprotein contains a large N-terminal extracellular region, multiple transmembrane domains, and a cytoplasmic C-tail. The protein functions as a regulator of calcium-permeable cation channels and intracellular calcium homoeostasis, is involved in cell–cell/matrix interactions, and modulates G-protein-coupled signal-transduction pathways. *PKD2* encodes a member of the polycystin protein family. PKD2 is a multi-pass membrane protein that acts as a calcium-permeable cation channel and is involved in calcium transport and calcium signaling in renal epithelial cells. Both polycystins modulate tubular morphogenesis [2].

Mutations in the *PKD1* gene are found in about 85% of ADPKD patients, mutations in the *PKD2* gene in about 15%, and mutations in the *GANAB* gene in approximately 1%. Approximately 10–15% of ADPKD patients do not have a documented family history of the disease, indicating a significant de novo mutation rate, and there are currently no known alternative *PKD*-causing mutations. Although the phenomenology of *PKD1* and *PKD2* mutations is identical, *PKD2* patients experience less severe disease, a less early development of hypertension and atherosclerosis, and probably fewer renal cysts compared to PKD1 patients. In contrast to *PKD1* patients, those with *GANAB* mutations experience greater comorbid hepatic illness, although a milder phenotype overall.

ADPKD is a significant cause of end-stage renal disease (ESRD), and is differentiated from autosomal dominant polycystic liver disease (ADPLD), which results in substantial PLD with pronounced liver involvement and minimal PKD [4,5,6]. Currently, there are eight genes, which have been associated with ADPKD (*PKD1* and *PKD2*), ADPLD (*PRKCSH*, *SEC63*, *LRP5*, *ALG8*, and *SEC61B*), or both (*GANAB*) [7]. In pediatrics, it is well known that autosomal recessive polycystic kidney disease (ARPKD) is prevalent, but recently, it has been suggested that a biallelic disease including at least one fragile ADPKD allele may result in a significant cause of symptomatic, very-early-onset (VEO) ADPKD, which is characterized by a glomerulocystic change in the kidney parenchyma [8,9,10]. In this chapter, we illustrate the severity of ADPKD and the ultimate fate to develop renal cell carcinoma (RCC).

## 2. ADPKD Severity and ESRD

The severity of disease in ADPKD is highly variable (Figure 1). We know that half of the affected individuals reach ESRD by approximately 60 years of age [11,12]. On the other side, less than 1% of individuals exhibit a VEO form of the disease, with a diagnosis made in utero or during early infancy [7]. Although subjects affected with ADPKD can live a normal lifespan without requiring a kidney transplant, numerous patients with ADPKD advance their disease by showing liver cysts. Particularly, cystic degeneration is more common as these patients become older [2]. ADPKD is genetically heterogeneous. There are two major genes, *PKD1* (16.p13.3; approximately 80% families) and *PKD2* (4p21; approximately 15%), and a rare third locus, *GANAB* (11q12.3; approximately 0.3%), which was discovered recently [3,10,11,12,13,14,15]. In the setting of ADPLD, *PRKCSH* (19p13.2; approximately 20%) and *SEC63* (6q21; approximately 15%) are the major genes. *LRP5* (11q13.2), *GANAB* (approximately 2%), *ALG8* (11q14.1; approximately 3%), and *SEC61B* (9q22.33; approximately 1%) have more recently been associated with ADPLD [3,5,13,14]. Multiple investigations have delineated the difference in the survival of kidneys and, ultimately, of the patients between PKD1 and PKD2 patients. In total, *PKD1* patients have a larger height-adjusted total kidney volume (HtTKV; an early measure of the severity of renal disease in ADPKD) and lower eGFR than *PKD2* patients. Moreover, the number of renal cysts is different in *PKD1* and *PKD2* patients. *PKD2*-gene-harboring patients exhibit fewer cysts than *PKD1*, despite the fact that the rate of growth of the complex renal parenchyma does not differ between the two groups. Occasionally, the severity of PLD can be the major clinical determinant in the phenotype of patients with ADPKD.

## 3. Pathology Workup

The association between ADPKD and RCC remains a hot topic because of the controversy linked to the distortion of the renal anatomic, imaging, and histologic architectural findings and the nonspecific cancer symptoms. The imaging-based diagnosis or follow-up may not be obvious. Thus, a thorough pathology workup is critical in all nephrectomy specimens with polycystic kidney disease and may have been overlooked in the past. Macroscopy or the grossing of the specimen is key. The specimen needs to be grossed with chessboard diagrams of the serial sections to avoid missing the very small foci of RCC. Figure 2 shows macroscopic imaging of the renal parenchyma of a pediatric patient with multifocal ADPKD-associated RCC. A histological examination of the renal parenchyma with cystic degeneration and solid areas is also critical. Multiple levels at 0.5 cm may be needed to confirm the presence of a neoplasm in a renal specimen of cystic renal disease. Multiple levels cut at 1 cm may not satisfactorily identify small cancer foci. Figure 3 shows the histopathology of a nephrectomy specimen from a pediatric patient with ADPKD with multifocal RCC, which is highlighted in the high magnification shown in Figure 4.

## 4. ADPKD-RCC Conundrum

The relationship between ADPKD and RCC remains nebulous and has never been fully confirmed, but there are two concepts that we need to have clear in a carcinogenetic process. We need to have an initiation factor and a promoting factor. The initiation factor may be linked to the genes associated with ADPKD and the promoting factor is highly likely to be the inflammation. Both ADPKD and RCC are relatively common in the common population and a few cases have been described in childhood [15,16,17,18,19,20,21,22,23,24,25,26,27,28,29,30,31,32,33,34,35,36,37,38,39,40].

There are a number of shared features between ADPKD and RCC, such as the development of kidney cysts, neoplastic cell proliferation, and a lack of effective treatments. They also share a compromised vasculature and the compensatory activation of angiogenesis by vascular endothelial growth factor (VEFG). Despite the lack of evidence linking ADPKD to an increased risk of RCC, J. Grantham has hypothesized that advanced renal disease might be considered “*neoplasia in disguise*” because of the multifocal neoplasia-like aberrantly expanding epithelial cells identified in the region of the side wall [22,41]. The difference between multifocal neoplasia and numerous tumors is that the former is thought to start with a single-cell clone and subsequently expand to become a multifocal process in just one organ [42]. Multifocal renal adenomas and frank adenocarcinomas are frequently preceded by polycystic kidney disease [43]. Cysts, adenomas, and carcinomas can share some morphologic and molecular characteristics. Similar to adenomas, cysts have swollen epithelial walls that compress and collapse the underlying parenchyma. The difference between adenomas and ADPKD cysts is that the latter are filled with cells and the former with fluid from glomerular filtrate. There may be some similarities between the stages of clear cell RCC, one of the most frequent variants of RCCs. It can start as ADPKD-like cysts in the nephron, progress to cystadenomas, and finally, become malignant with progressive invasive features [44,45].

There are similarities between ADPKD cysts and clear-cell RCC on a molecular level [22]. Although individual mutations do not induce RCC, the conditional deletion of the *Vhl* and *Pbrm1* genes, which are also involved in carcinogenesis, in the cells of the murine renal epithelium can lead to preneoplastic polycystic kidney disease and, ultimately, RCC [44]. By utilizing Cre/lox recombination in the Ksp-Cre line, which carries the *Ksp*-cadherin kidney-specific promoter, *VhlF*/*FPbrm1 F*/*FKsp-Cre* mice were produced. These mice are a well-used mouse model of RCC [8]. Approximately half of the tumors in *VhlF*/*FPbrm1 F*/*FKsp-Cre* mice develop multifocal, clear-cell-type RCC by the time they reach 10 months of age [44]. There was an increased mortality rate, greater serum creatinine, and the presence of preneoplastic PKD-type cysts in the tubules and glomeruli by six months of age in these rodents, just like we can demonstrate in patients harboring Von Hippel Lindau disease [46,47].

RCC, ARPKD, and ADPKD all have aberrant vascular appearances [48]. By reabsorbing the glomerular filtrate, the renal cortex’s peritubular reticular capillaries control the fluid balance. As a result of ADPKD, the blood remodeling of vasculature is believed to play a role in the advancement of the disease [49]. The peritubular capillaries enlarge, become convoluted, and highlight a spiral appearance in ADPKD patients and *Pkd1^nl/nl^* mice [50]. Using experimental animals, mice lacking *Pkd1* also exhibit fewer branching and segmented vessels than the wild type [51]. Additionally, the lymphatic capillaries, which are responsible for draining the interstitial fluid around the organs, undergo remodeling, have several abnormalities, and have reduced branching [52]. Researchers found that the early mortality of the *Pkd1*^−/−^, *Pkd1^nl/n^*, *Pkd1^RC/RC^*, and *Pkd2^−/−^* mice is caused by blood accumulation in the lymph sacs, which is caused by a compromised lymphatic function [53]. Cyst enlargement, like tumor growth, necessitates a change in the vasculature to supply metabolites that support angiogenesis, a temporary repair strategy that starts by producing new capillaries from the current vasculature to deliver blood to the healed tissue. Vascular endothelial growth factor (VEGF)-A is one of the main growth factors that induce angiogenesis [54,55,56]. It binds to the receptor VEGFR-2 and encourages the proliferation, differentiation, migration, and survival of blood endothelial cells [57]. Increased angiogenesis may facilitate cyst growth by VEFG-A, which is expressed in both large cysts and cortical tubules [48]. The activation of VEFG-C and its receptor VEGFR-3 also enhances cell proliferation, differentiation, migration, and survival in lymphatic endothelial cells [53], although the exact mechanisms are still covered with mystery. It has been suggested that polycystic kidney disease can be improved by activating VEFG-C, which increases lymphatic channels and so facilitates the evacuation of extra fluids and inflammatory cells from the renal interstitial space [50].

In cancer, VEFG promotes long-term angiogenesis, which results in the formation of aberrant, highly permeable, and disorderly arranged blood vessels [58]. An aberrant increase in the capsular vascular supply with the outflow through the ovarian or testicular veins is caused by *VEFG* overexpression in the clear-cell type of RCC, which in turn causes complicated neovascularization [59]. Despite this enhanced supply, blood perfusion within the neoplasm seems to be lower than in healthy tumor-surrounding renal tissues [59]. It has also been suggested that this could be due to the fact that this type of RCC often forms a network of tiny sinusoidal blood capillaries [60].

Many chromosomal abnormalities are seen in RCC, particularly in advanced stages, which is a sign of genetic instability [61]. A protein typically formed by interactions with Elongin B and C, Cul2, and Rbx1 is encoded by the *VHL* gene [62]. VHL haploinsufficiency, or “first hit”, is not sufficient to generate cancer; however, it shifts the metabolism by activating some of the Warburg effect factors and some of the mediators of the glutamine reductive metabolism, specifically placing the cells on the path to transformation [22]. Prolyl hydroxylase domain proteins hydroxylate hypoxia-inducible factor (HIF) in normoxia, increasing its affinity for VHL [54,63]. Tubular epithelial cells, glomerular and peritubular cells, and the EPO-producing kidneys all contain different isoforms of the heat shock factor. Typically, HIF-1 is targeted for proteasomal degradation by the VHL-containing ubiquitin E3 ligase complex [63]. A stable and constitutively produced HIF-1 component is formed when hypoxic stress stabilizes and translocates HIF-2 and HIF-1 to the nucleus, forming a heterodimer [63]. The cell response to oxygen deprivation is initiated when transcriptional coactivators such as p300/CBP and the HIF heterodimers bind to hypoxia response elements (HREs) to increase the transcription of hypoxia target genes (e.g., VEFG, glucose-transporter 1—GLUT1, EPO) [22]. The PDGF and epidermal growth factor receptor (EGFR) genes seem to be regulated in different ways, and they do not contain HREs. However, both EGFR and PDGF are implicated in the hypoxia response and promote vascularization in nearby endothelial cells [64]. These variables are believed to work together to alter the metabolism during cancer development.

Somatic mutations generate a “second hit” that leads to cancer in the long run or changes to the epigenome that render the second copy of VHL inactive [22,65]. A number of genes have been found to be related with cancer, including *PBRM1*, *BAP1*, *SET domain-containing 2*, Lysine demethylase (*KDM5C*), *mTOR*, *PTEN*, PI3K catalytic subunit, *TP53*, and Elongation Factor B (*TCEB1*) [66]. After the loss of function for the *VHL* and *PBRM1* genes, cellular proliferation and mTOR activation are hallmarks of the clear-cell type of RCC tumors in both mice and humans [44]. Even though human and mouse genomes are often very similar, there is a noticeable difference in the organization of the homologous RCC genes between the two species [28]. Mice models are thus unable to accurately portray the genetic changes that cause RCC in humans. The PI3K/Akt/mTOR signal transduction pathway, HIF-1 and HIF-2, VEFG, EGFR, CA-IX, GLUT transporters, TGF-, Notch, and transforming growth factor (TGF) are additional factors that contribute to the formation of RCC [67,68]. It has also been demonstrated that the VHL/HIF and PI3K/AKT pathways communicate with one another as part of a complex signaling network that contributes to the advancement of carcinogenesis [69]. Tumor cell migration, angiogenesis, cell fitness, and their formation are all impacted by these cascades.

Hypoxia, which activates HIF-signaling in reaction to oxygen deprivation, is a common outcome of both RCC and ADPKD. The development of the disease may be influenced by primary cilia and changes in the cell metabolism, as suggested above. The diverse impacts of non-coding RNAs suggest that they may influence RCC carcinogenesis and ADPKD as well. Several pathways that are involved in RCC and ADPKD are quite conserved in *Drosophila* and this species may be key to further characterizing the relationship between ADPKD and cancer [22].

Finally, other findings may disguise the ADPKD-RCC tendency or sequence. One of the most frequent issues is the lack of clear morphological clues in identifying the progression from inflammatory atypia to neoplastic atypia (Figure 5).

They include benign lesions, bilaterality, and multifocality. The correct and regular workup of nephrectomy specimens revealed an increased number of papillary adenomas, which are precursors of papillary RCC, malignant tumors in both nephrectomy specimens, and multifocality, i.e., the presence of more than one or two RCCs. On the other side, sarcomatoid neoplasms are inconstantly found. However, sarcomatoid RCC is a complex and complicated neoplasm, which may be represented, at least in some settings, as a degenerated RCC with increased atypia and aggressiveness. Some radiological studies have been identified as fallacious because architectural distortion, intracystic bleeding, and infection may prevent a proper assessment of the imaging. We fully advocate for a long-term follow-up of ADPKD individuals, even though in some centers, prophylactic bilateral nephrectomy is carried out early to avoid the development of cancers [38,70]. On the other side, newer imaging technologies are gifted to envisage accurately small tumors and complex cystic structures with a nonsignificant risk of metastasis. Favoring a bilateral nephrectomy should be balanced with the higher morbidity rate and the poorer quality of life of the patients, who will need adequate targeting for the anuria and lifelong supplementation of erythropoietin [38].

## 5. Role of Inflammation as Promoter in ADPKD-Related RCC

NLRP3 is an immunologically key 115 kDa protein which has been studied by our group [71,72,73]. NLRP3 consists of three domains. There is a central oligomerization domain also known as NOD, a nucleotide-binding domain, a C-terminal LRR domain, which stands for leucine-rich repeat, and a pyrin N-terminal effector PYD domain [74,75]. The PYD domain can recruit the Apoptosis-associated Speck-like protein containing a CARD or Caspase recruitment domain ASC adapter. It has been disclosed that NLRP3 is an amazing cytosolic stress sensor. The NLRP3 has been the subject of studies in our group and is an underevaluated pathway for therapeutical options. NLRP3 can trigger a pro-inflammatory signaling pathway involved in the innate immune response [76,77,78]. Following NLRP3 activation, there is an assembling of the multiprotein signaling complex in the cytosol. It is called the inflammasome due to powerful complex and its interaction with NEK7 (NIMA-related kinase 7). This protein belongs to NIMA-related kinases. It is an essential mediator of NLRP3 activation downstream of the potassium efflux [79]. The mammalian NEK group belongs to a serine/threonine kinase named NEK1–NEK11, which overviews several aspects of non-mitotic and mitotic properties [80]. After that, there is the recruiting of the ASC adapter via its PYD domain. It is triggered through its CARD domain.

Ultrastructurally, the inflammasome is visualized as “specks”, which are micrometric particles [81,82]. Both steps are critical for the activation of the NLRP3 inflammasome. The first is the priming signal, which activates the nuclear factor kappa-B (NF-κB) pathway. The proper activation or second signal induces the appropriate assembly of the inflammasome [83]. The NLRP3 inflammasome relates to incorporating cellular stress signals (e.g., potassium efflux exposure to microbial toxins, crystals, lysosomal rupture, and mitochondrial dysfunction). Raptis et al. (2020) studied 26 ADPKD patients with some impairment of the renal function and 26 age- and sex-matched controls [84]. They found that ADPKD patients disclosed higher NLRP3 levels than the controls. These data were also associated with increased levels of copeptin, a surrogate marker of arginine vasopressin release, and suPAR (soluble urokinase-type plasminogen activator receptor) levels, which were also increased compared with the controls. These findings support the role of the NLRP3 inflammasome in the progressive degeneration of cystogenesis and may support further studies investigating the NLRP3 inflammasome in carcinogenesis. It is interesting to speculate that if the NLRP3 inflammasome is targeted early, we may avoid the promotion of a benign tumor to malignancy or the onset of bilaterality and/or multifocality in patients with ADPKD.

The dysregulation of the epithelial–stromal interaction has been implicated in the pathogenesis of a variety of diseases, including cancer [85]. Fibrosis and degenerative changes develop secondarily to cystic changes, postulating that if an epithelial abnormality takes place, the stroma interaction may have a complex interaction with several epithelial structures and the interstitium (Figure 6). Indeed, fibrosis is a hallmark of cancer. Up to one fifth of cancers are linked to chronic-inflammation-related fibrosis (either from infectious or autoimmune etiologies) including esophageal, gastric, colonic, hepatocellular, pancreatic, head and neck, and cervix and vulvar cancers [86].

Tumors are characterized by extracellular matrix (ECM) deposition, remodeling, and cross-linking that drive fibrosis to stiffen the stroma and promote malignancy. The stiffened stroma enhances tumor cell growth, survival, and migration and drives a mesenchymal transition. A stiff ECM also induces angiogenesis and hypoxia and compromises anti-tumor immunity. Not surprisingly, tumor aggression and poor patient prognosis correlate with the degree of tissue fibrosis and the level of stromal stiffness. In their review, Piersma and colleagues discuss the reciprocal interplay between tumor cells, cancer-associated fibroblasts, immune cells, and ECM stiffness in malignant transformation and cancer aggression [87].

## 6. “Disguising” Epidemiology

In Europe, the annual incidence of RCC is about 5–20 in 100,000 male residents, while it is slightly lower in female residents with a rate of about 2–11 per 100,000 [37,88]. Autopsies are paramount and a key metric of quality in a hospital. The prevalence of ADPKD in autopsy series is probably about 0.5–1%, but the association of ADPKD with RCC has been often dismissed as coincidental in plenary clinical meetings, but the true rates may be quite different.

Epidemiologic studies have not been qualified as critical because of the number of confounding factors, including ESRD, which is related to the natural evolution of both ADPKD and RCC. ESRD is a well-known risk factor in evaluating the carcinogenetic potential of acquired renal cystic disease [7]. It seems that immunosuppression does not play a major role in carcinogenesis, but analgesic nephropathy and the acquisition of renal cysts in a patient without a genetic predisposition have been considered as dominant risk factors. The controversial hypothesis or theory that the native kidney may have incidental foci of pre-carcinoma that may develop to RCC in the event of prolonged immunosuppression, which is obviously lifelong, is debatable. We suggest that RCC prevalence in ADPKD individuals is possibly underestimated currently. It may be due to the decreased number of autopsies performed in a clinical or academic setting or due to the poor resolution of the imaging analysis or the presence of small foci that may be undetectable. In 2002, Denton et al. studied 260 native nephrectomies completed in ESRD patients who underwent transplantation [88]. They reported an RCC prevalence rate of 4.2%, which is approximately 1 in 20 patients affected with ADPKD. Astoundingly, there was an outcry in the medical community and none, at least non-pathologists, advocated to increase the accuracy of the quality in medicine with a compulsory autopsy as it is in some countries. If performed routinely, high-resolution imaging studies and full autopsies can probably be able to correctly estimate the increased susceptibility to RCC in patients harboring ADPKD at an early stage.

Several writers in the second half of the twentieth century highlighted the autopsy’s function as a quality assurance tool, drawing attention to the aforementioned distinctions between antemortem and postmortem diagnoses, while the aspect of “understanding the pathogenesis of disease and detecting new disease entities” largely fell into the background. Autopsies are still seen as useful in medical education, for example, as a source of case studies for problem-based learning. Medical students review anatomical concepts and learn about clinicopathological relationships through autopsy cases. Moreover, autopsies provide doubtless some groundwork for thinking about quality management in healthcare. The examination of the precision of diagnostic imaging, including computed tomography (CT), nuclear magnetic resonance (NMR), and positron emission tomography (PET) scans, is one aspect of the autopsy’s function in quality assurance. Novel medication and surgical procedure efficacy and safety assessments, as well as genetic engineering, are further areas of quality assurance that need to be tackled in this decade [89].

In Austria, there is no requirement for the consent of the parents or relatives for a clinical autopsy performed in a medical institution § 25 KAKuG Leichenöffnung (*Obduktion*) (Krankenanstalten und Kuranstaltengesetz). [(1) The corpses of deceased patients in public hospitals are to be autopsied if the autopsy has been ordered by sanitary or criminal prosecutors, or if it is necessary for public or scientific interests, in the situation of diagnostic uncertainty of the case, or after surgical intervention. (2) If none of the cases mentioned in point 1 is present and the deceased has not consented to an autopsy during his lifetime, an autopsy may only be carried out with the consent of the next of kin. (3) A copy of the medical history shall be recorded for each autopsy and kept in accordance with § 10, 1 Z. 3. In the past, § 25 KAKuG was labeled as § 25 KAG (Krankenanstaltengesetz).] However, in specific cases, a limited autopsy or a postmortem biopsy of one or two organs with a closed body is performed in lieu of a full autopsy to accommodate the wishes of family and relatives. In Austria, this autopsy procedure allows institutions to keep the autopsy rate to a higher level than the level of other neighboring Western countries, permitting high standards of quality assurance and quality controls in most of the Austrian healthcare institutions [90,91].

## 7. Therapeutical Options

The principal role of the polycystin protein is to regulate transmembrane calcium transport, which in turn controls the formation of the kidney’s and other organs’ vascular and tubular networks [92]. Multiple growth factors and secondary messengers, such as cAMP, adenosine, insulin growth factor, and epidermal growth factor, work together to accomplish this. During the cysts’ proliferative stage, ERK and mTOR regulate and govern several processes [93,94]. An increase in cyclic adenosine monophosphate (cAMP) is a mediator of the epithelial cell proliferation that leads to larger cysts and fluid buildup in response to the anti-diuretic hormone (ADH) [92,95]. When cysts grow big enough, they can function independently. The therapy mechanism for ADPKD is based on drugs that act on V2 ADH receptors, resulting in a decrease in cAMP levels [94].

In order to decrease cAMP levels and delay the start and advancement of ADPKD, tolvaptan was created as a specific V2 receptor antagonist (Figure 7). When compared to natural ADH, tolvaptan’s affinity for the V2 receptor is two times higher, and when compared to the V1A receptor, it is 29 times stronger [94]. It treats ADPKD and is the only medicine of its kind to have FDA approval [96].

Tolvaptan, a selective V2 receptor antagonist, is currently FDA-approved for use only in adults [94]. Tolvaptan has a 40% oral bioavailability and is quickly absorbed once it reaches the bloodstream. According to the research, the effects of meals on tolvaptan are small and dose-dependent, according to the research [94]. Following absorption, the steady-state concentration reaches a plateau at 300 mg. Cytochrome p450 is responsible for the hepatic metabolism of tolvaptan after it binds to plasma proteins. Tolvaptan is eliminated from the body through the feces and has a half-life of 12 h. Few studies have examined the pharmacokinetic and pharmacodynamic features of tolvaptan. Most of the research indicates that patients with creatinine levels below 10 mg/dL should not be administered tolvaptan. Additionally, when other medications utilize the Cytochrome p450 (CYP 3A4) pathway concurrently, tolvaptan is linked to drug–drug interactions. Due to their impact on the tolvaptan hepatic metabolism, CYP 3A4 inducers and inhibitors should not be used when tolvaptan is being administered.

The pathophysiology of ADPKD is affected by several physiological alterations brought about by tolvaptan’s V2 receptor antagonism in the renal collecting ducts on reaching the kidney level [12,94,97,98,99]. It raises the serum Na+ concentration and lowers the urine osmolality, and it induces free water excretion, leading to net body fluid loss [96]. Another benefit of V2 antagonism is a the slowing ADPKD progression due to a reduction in cyst proliferation caused by cAMP [9,12,98,100].

Only adults can take tolvaptan according to the current United States Food and Drugs Administration (FDA) guidelines. Tolvaptan’s effectiveness has been the subject of twenty-two trials spanning 2011–2021 [94,97]. The results from these trials demonstrated that tolvaptan reduced the rate of decline in the estimated glomerular filtration rate and the rise in the total kidney volume in ADPKD patients. In the two biggest clinical trials conducted on individuals with ADPKD, REPRISE, and TEMPO 3:4, the safety and effectiveness of tolvaptan were examined. When compared to a placebo, each of these treatments slowed the kidneys’ deterioration. For REPRISE, the change in eGFR from the baseline to post-treatment was 1.3 mL/min/1.73, while for TEMPO 3:4, it was 1 mL/min/1.73. The total renal capacity decreased by 49% on average between the pre- and post-treatment measurements in the TEMPO 3:4 trial [94].

Patients with ADPKD between the ages of 18 and 50 were the focus of the randomized clinical trial known as TEMPO 3:4. Over the course of three years, 1445 participants with renal volumes greater than 750 mL and creatinine clearances greater than 60 mL/min were given either a placebo or tolvaptan. Tolvaptan was started with a 45 mg morning dose and a 15 mg afternoon dose, and the dosage gradually increased. The percentage change in the total renal volume was the main outcome [94]. The total kidney volume changed by 9.6% with tolvaptan and 18.8% with the placebo. Reduced rates of renal function worsening (as indicated by changes in eGFR and increases in serum creatinine) and fewer episodes of kidney pain were observed in patients treated with tolvaptan. On average, blood creatinine levels increased by 0.16 mg/dL after tolvaptan treatment, compared to 0.23 mg/dL after the placebo. In the tolvaptan group, the annual eGFR slope change was 2.72 mL/min/1.73 m^2^, while in the placebo group, it was 3.70 mL/min/1.73 m^2^. There was a marked decrease in albuminuria in patients who were given tolvaptan. By tracking 871 patients who finished TEMPO 3:4, TEMPO 4:4 investigated tolvaptan’s long-term effects. For the following twenty-four hours, patients took 45, 60, or 90 mg first thing in the morning and either 15 or 30 mg again nine hours later. After 24 months, patients on tolvaptan had a decrease of 29.9% in their total kidney volume from TEMPO 3:4 to TEMPO 4:4, while patients taking the placebo had a decrease of 31.6% (*p* was not significant). During TEMPO 4:4, there was a 3.26% annual decrease in eGFR with early treatment during TEMPO 3:4 and a 3.14% annual decrease with the delayed treatment. No significant changes in eGFR were observed during long-term tolvaptan therapy. Treatment with tolvaptan slowed the growth of the total kidney volume and restored renal function in the near term, but in the long run, the effects were maintained.

Patients in the REPRISE study ranged in age from 18 to 55 years old and had an estimated glomerular filtration rate (eGFR) of 25 to 65 mL/min/1.73 m^2^ at the baseline. Patients in the 56 to 65 age group had an eGFR of 25 to 44 mL/min/1.73 m^2^ and an eGFR drop of more than 2 mL/min/1.73 m^2^ per year. A 12-month trial of tolvaptan or a placebo was conducted with participants taking 60 or 90 mg of tolvaptan first thing in the morning and 30 mg in the afternoon. The data demonstrated that at the 1-year follow-up, the average decrease in eGFR for patients treated with tolvaptan was −2.34 mL/min/1.73 m^2^, while for patients given the placebo it was −3.61 mL/min/1.73 m^2^. With the exception of non-whites, those older than 55, and those with early CKD, all categories showed the changes.

There have been several investigations into the safety and effectiveness of tolvaptan in ADPKD patients, but so far, it is currently FDA-approved for use only in adults, as indicated above [94]. To the best of our knowledge, tolvaptan (JYNARQUE^TM^) is not currently approved for use in children. While it is FDA-approved for adults at risk of rapidly progressing ADPKD, studies in pediatric patients are ongoing, but no pediatric approval exists. Researchers at the National Institutes of Health (NIH) (.gov) are conducting a phase 3, 1-year, randomized, double-blind, placebo-controlled, multicenter trial in participants with ADPKD aged 4–17 years to evaluate the pharmacodynamics, safety, and efficacy of tolvaptan in children and adolescents with ADPKD [97].

## 8. Conclusions

In conclusion, ADPKD may involve cancer, and clinical oncogenetic studies reveal that at least one patient with ADPKD out of fifteen harboring this genetic disease will develop RCC, as highlighted in several studies (e.g., [38]). The risk increases to 12% or even higher in ADPKD patients after one year of hemodialysis or renal transplantation [38]. The role of the NLRP3 inflammasome in ADPKD and ADPKD-RCC is intriguing and may be targeted in future investigations to properly identify pathways, which may reveal the rate of oncogenesis in ADPKD. Autopsies as quality metrics in healthcare institutions have become a rarity and it is presumable to hypothesize that if autopsy rates were higher than current rates, the RCC linked to ADPKD would probably be higher. Moreover, we suggest that that investigations involving multiple centers should be planned in the future and such studies should be randomized. Clinical investigative studies should be adjusted to the duration of hemodialysis and organ transplantation, as well as to other risk factors, which may be varied from country to country worldwide. We suggest that a bilateral nephrectomy should be carried out in ADPKD patients with complex cysts or when the imaging is not convincing in excluding a life-threatening neoplastic conversion.

## Figures and Tables

**Figure 1 ijms-26-03965-f001:**
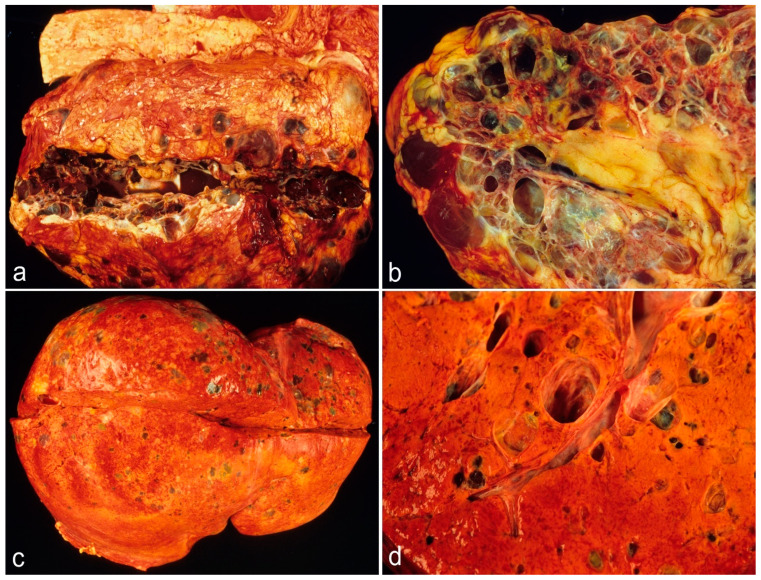
Adult ADPKD. This panel of four gross photographs shows Autosomal Dominant Polycystic Kidney Disease (ADPKD) in a young man harboring a mutation of the *PKD1* gene at autopsy. Multiple cysts of the kidneys (right kidney, displayed, left kidney not displayed), some of them hemorrhagic, are shown (**a**,**b**). Histological examination of the kidneys did not show any evidence of carcinoma. Interestingly, this patient showed early severe atherosclerosis of the blood vessels, which is particularly demonstrated in (**a**) at the level of the thoracic and lumbar aorta. This patient showed also multiple liver cysts, as demonstrated in (**c**,**d**), but obviously the number of the cysts of the liver is inferior to the number of cysts of the kidneys. The gross photographs come from the personal archive of Dr. Sergi.

**Figure 2 ijms-26-03965-f002:**
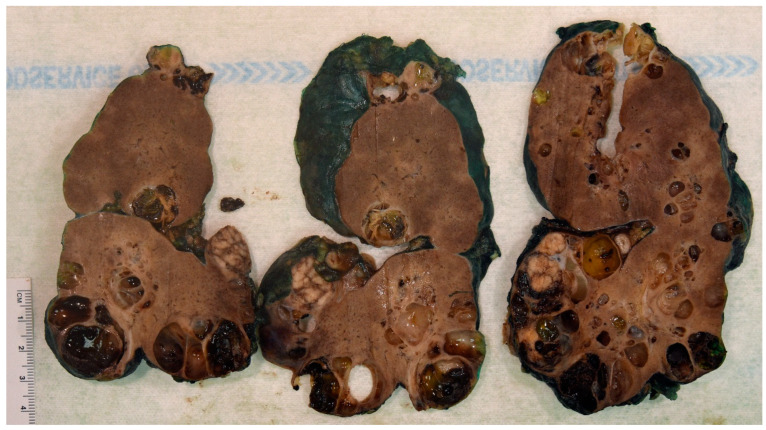
ADPKD-RCC in a pediatric patient. Gross photographs of the kidney specimen from a child harboring ADPKD showing several cysts, some of them are hemorrhagic, and harboring nodules of increased consistency, which have been demonstrated to be harboring neoplastic features. The gross photographs come from the personal archive of Dr. Sergi.

**Figure 3 ijms-26-03965-f003:**
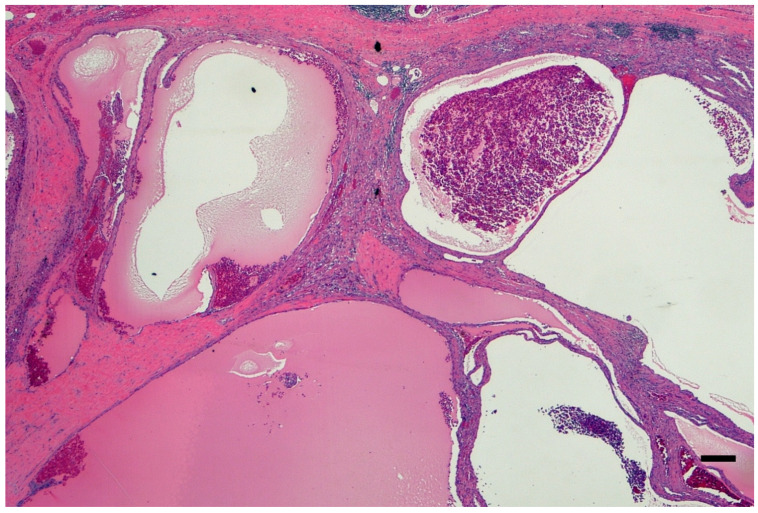
Low-power microphotograph showing numerous cysts with fluid and erythrocytes in the lumen as well as lack of embryological dysplastic features (e.g., cartilage or primitive ducts) (Hematoxylin and Eosin Staining, ×20 original magnification, bar, 500 μm). The microphotograph comes from the personal archive of Dr. Sergi.

**Figure 4 ijms-26-03965-f004:**
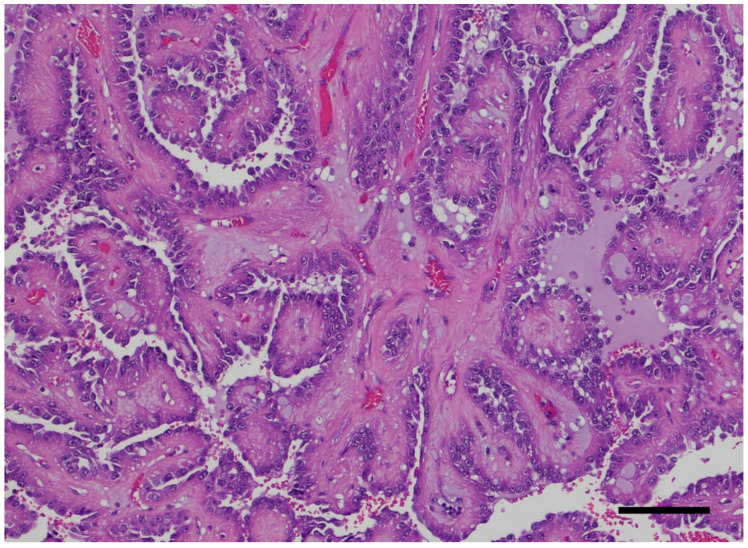
High-power microphotograph of a multifocal renal cell carcinoma showing a mainly papillary growth pattern. The nuclear morphology is Fuhrman grade 2 (Hematoxylin and Eosin Staining, ×100 original magnification, bar: 100 μm). The microphotograph comes from the personal archive of Dr. Sergi.

**Figure 5 ijms-26-03965-f005:**
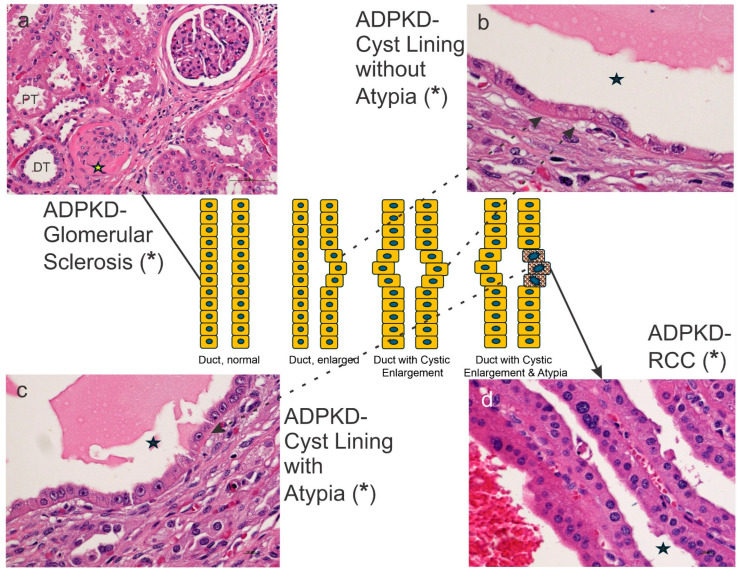
ADPKD progression to ADPKD-RCC. From (**a**–**d**), ADPKD glomerular sclerosis advances to ADPKD harboring cysts without atypia (**b**), through ADPKD cysts exhibiting atypical features (**c**), to classic ADPKD-cancer (RCC, renal cell carcinoma). The yellow asterisk in (**a**) shows a glomerular sclerosis, while the asterisks in (**b**,**c**) highlight the lumen of the cyst, while neoplastic cancer cells are seen (asterisk) in the papillae of this ADPKD-RCC. In the center, there is the progression from normal duct, through enlarged duct, to duct with cystic enlargement and duct with cystic enlargement and atypia. DT, distal tubule; PT, proximal tubule. The diagram has been designed by Dr. C. Sergi using Microsoft Power Point, Windows 11, and microphotographs that come from his personal archive.

**Figure 6 ijms-26-03965-f006:**
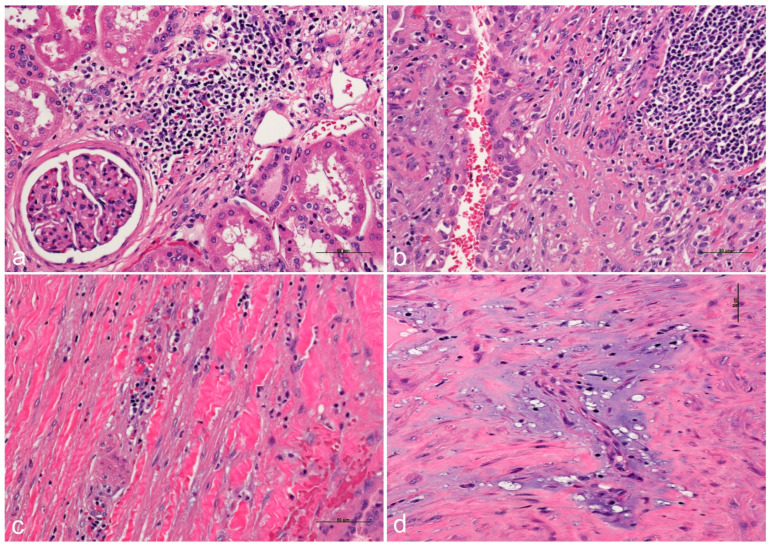
Inflammation in ADPKD-RCC. Inflammation with mononuclear cells is seen in periglomerular regions (**a**), destroying the interstitial renal parenchyma in (**b**), percolating collagen fibers (**c**), and as scattered inflammatory cells in a myxoid or myxoid-like background (**d**) (Hematoxylin and Eosin Staining, ×100 original magnification; bar values are disclosed in the microphotographs). All microphotographs come from the personal archive of Dr. C. M. Sergi.

**Figure 7 ijms-26-03965-f007:**
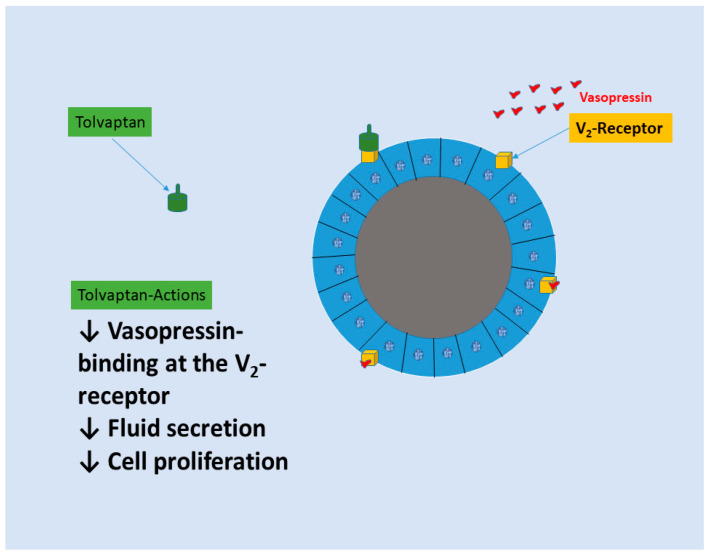
Tolvaptan actions (see text for details). The photo has been realized by the first author (CMS) using Microsoft PowerPoint, Windows 11.

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
