# Peer review of "Autosomal Dominant Polycystic Kidney Disease-Related Multifocal Renal Cell Carcinoma: A Narrative Iconographic Review"

_ijms, 2025, doi:10.3390/ijms26093965_

Round 1
Reviewer 1 Report
Comments and Suggestions for Authors
This manuscript covers autosomal dominant polycystic kidney disease and renal cell carcinoma. However, the manuscript contains several significant shortcomings that necessitate major revisions.
The lack of clearance is one of the most outstanding drawbacks of this manuscript. While the title implies that this manuscript will focus on the connection or correlation between ADPKD and RCC, several other topics are discussed in the manuscript, which, while making the manuscript more comprehensive, do not seem relevant and should not be included. Such examples include the discussion about autosomal dominant polycystic liver diseases.
Another outstanding problem of the manuscript is the discrepancies between abstract, main text and conclusions. For example, the abstract does not comprehensively summarize the main text. In addition, in ‘Conclusions’ section, the statement about bilateral nephrectomy is not supported in main text at all. There are more examples in the main text.
In addition, in section 5, the author should raise the examples of inflammation in ADPKD-related RCC. The references and examples of NLRP3 are not sufficient in demonstrating the role of inflammation as promoter in ADPKD-related RCC.
There are also some formatting problems in the main text, like the incorrect references number in section 2 ‘ADPKD Severity and ESRD’.
Therefore, the author should carefully revise the manuscript to ensure the clarity and scientific accuracy of the statement.
Author Response
This manuscript covers autosomal dominant polycystic kidney disease and renal cell carcinoma. However, the manuscript contains several significant shortcomings that necessitate major revisions.
The lack of clearance is one of the most outstanding drawbacks of this manuscript. While the title implies that this manuscript will focus on the connection or correlation between ADPKD and RCC, several other topics are discussed in the manuscript, which, while making the manuscript more comprehensive, do not seem relevant and should not be included. Such examples include the discussion about autosomal dominant polycystic liver diseases.
Another outstanding problem of the manuscript is the discrepancies between abstract, main text and conclusions. For example, the abstract does not comprehensively summarize the main text. In addition, in ‘Conclusions’ section, the statement about bilateral nephrectomy is not supported in main text at all. There are more examples in the main text.
In addition, in section 5, the author should raise the examples of inflammation in ADPKD-related RCC. The references and examples of NLRP3 are not sufficient in demonstrating the role of inflammation as promoter in ADPKD-related RCC.
There are also some formatting problems in the main text, like the incorrect references number in section 2 ‘ADPKD Severity and ESRD’.
Therefore, the author should carefully revise the manuscript to ensure the clarity and scientific accuracy of the statement.
Thank you for your comments and suggestions. The manuscript was thoroughly reviewed, more information on the ADPKD-RCC sequence was added, several more photographs added, details on the morphological change of the cytology in the sequence ADPKD-RCC sequence, and a full new section on Tolvaptan. The references are now 100 and I have two senior authors (Dr. Guerra, Chief of Pediatric Urology) at my hospital and Dr. Hager, Chief of Pediatric Surgery and mentor at the Medical University of Innsbruck.
The section on NRLP3 inflammasome was also expanded.

Reviewer 2 Report
Comments and Suggestions for Authors
Major comments:
i) Page 4: “We need to have an initiation factor and a promoting factor. The initiation factor may be linked to the genes associated with ADPKD and the promoting factor is highly likely the inflammation.”; this sentence is a rather naïve and simplistic explanation for a carcinogenic process. The author needs to provide a more robust explanation.
ii) Page 4: “Autopsy is paramount and a key metric of quality in a hospital.”; the sentence is somewhat hanging in the manuscript.
iii) Page 4: “It may be due to the decrease number of autopsies performed in a clinical or academic setting or due to imaging analysis not properly performed due to lower skills in specialization schools.”; this sentence needs to be justified by the author. What does the author really mean with this? Are specialization schools not really specialized?
iv) Page 5: “as it is in some countries”; to which country is the author referring to and making comparisons?
Minor comments:
i) Page 1: “have been subsequently added complexity”; what is the term “been” doing in this sentence?
ii) Page 1: “different classification that are supported”; classifications?
iii) Page 1: “but it presently seems”; this segment needs correction.
iv) Page 1: “his highly inconstant.1”; Should it be variable instead of inconstant? And the number 1 is just a mistake? Or a reference number?
v) Page 2: “these patients become older7,8.”; are 7 and 8 references? If so why not having them as [7,8]?
vi) Page 2: “discovered last year.3,10–15”; change for [3,10-15].
vii) Page 2: “ADPLD.15–20”; change for [15-20]; and many other examples in the text. Homogenise the way references are cited in the manuscript.
viii) Page 5: “In the cytoplasm.”; what does this mean?
Author Response
Major comments:
- i)Page 4: “We need to have an initiation factor and a promoting factor. The initiation factor may be linked to the genes associated with ADPKD and the promoting factor is highly likely the inflammation.”; this sentence is a rather naïve and simplistic explanation for a carcinogenic process. The author needs to provide a more robust explanation.
- ii)Page 4: “Autopsy is paramount and a key metric of quality in a hospital.”; the sentence is somewhat hanging in the manuscript.
iii) Page 4: “It may be due to the decrease number of autopsies performed in a clinical or academic setting or due to imaging analysis not properly performed due to lower skills in specialization schools.”; this sentence needs to be justified by the author. What does the author really mean with this? Are specialization schools not really specialized?
- iv) Page 5: “as it is in some countries”; to which country is the author referring to and making comparisons?
Minor comments:
- i)Page 1: “have been subsequently added complexity”; what is the term “been” doing in this sentence?
- ii)Page 1: “different classification that are supported”; classifications?
iii) Page 1: “but it presently seems”; this segment needs correction.
- iv)Page 1: “his highly inconstant.1”; Should it be variable instead of inconstant? And the number 1 is just a mistake? Or a reference number?
- v)Page 2: “these patients become older7,8.”; are 7 and 8 references? If so why not having them as [7,8]?
- vi)Page 2: “discovered last year.3,10–15”; change for [3,10-15].
vii) Page 2: “ADPLD.15–20”; change for [15-20]; and many other examples in the text. Homogenise the way references are cited in the manuscript.
viii) Page 5: “In the cytoplasm.”; what does this mean?
Tha manuscript was thoroughly revised with more photographs, more details on the sequence ADPKD-RCC, and more information on genetics and pathogenesis as well as a fully new section on Tolvaptan.
All comments, major and minor, have been addressed.
There are two more authors, who are the chief of pediatric urology at my hospital and my former chief of pediatric surgery at the Medical University of Innsbruck.

Reviewer 3 Report
Comments and Suggestions for Authors
ADPKD is one of the most common congenital diseases in CKD population and many studies have been conducted on this topic over the years. Patients with ADPKD present a high risk of cystic neoplasia transformation, but RCC prevalence can be up to 8.73% according to a recent study. Therefore, the present study can highlight important features in the management of these patients and not to overlook the possible development of this type of cancer. But considering the title "iconographic review", more details should be included: iconographic means representing something by pictures or diagrams, therefore, more figures should be presented regarding not only the general and HP aspect of the kidneys, but also related to the inflammation underline mechanisms in order to better visualise the magnitude of the problem. It would be important to discuss also, in a different section of the article, the available treatment options to this group of patients (i.e., the selective antagonist that blocks arginine vasopressin from binding to V2 receptors) and the influence in the onset of RCC (including also figures when presenting this information, as it is an iconographic review) - some aspects were discussed, but not sufficiently. Considering that the type of the article is a review, more data should be provided related to this complex topic, including images, as well. In addition, more data regarding your experience regarding this topic should be included, as you mentioned "We and others consider that the RCC prevalence in ADPKD individuals is probably underestimated".
Author Response
ADPKD is one of the most common congenital diseases in CKD population and many studies have been conducted on this topic over the years. Patients with ADPKD present a high risk of cystic neoplasia transformation, but RCC prevalence can be up to 8.73% according to a recent study. Therefore, the present study can highlight important features in the management of these patients and not to overlook the possible development of this type of cancer. But considering the title "iconographic review", more details should be included: iconographic means representing something by pictures or diagrams, therefore, more figures should be presented regarding not only the general and HP aspect of the kidneys, but also related to the inflammation underline mechanisms in order to better visualise the magnitude of the problem. It would be important to discuss also, in a different section of the article, the available treatment options to this group of patients (i.e., the selective antagonist that blocks arginine vasopressin from binding to V2 receptors) and the influence in the onset of RCC (including also figures when presenting this information, as it is an iconographic review) - some aspects were discussed, but not sufficiently. Considering that the type of the article is a review, more data should be provided related to this complex topic, including images, as well. In addition, more data regarding your experience regarding this topic should be included, as you mentioned "We and others consider that the RCC prevalence in ADPKD individuals is probably underestimated".
Thank you very much for your comments and suggestions. The manuscript was fully revised with more photographs, sections, and data. Moreover, a full section on Tolvaptan is provided.

Round 2
Reviewer 3 Report
Comments and Suggestions for Authors
You have answered to all the questions and the article has been clearly improved. In this form, the article better describes ADPKD key features, and most importantly the association with RCC.